# Temporal Dynamics of High-Density Lipoprotein Proteome in Diet-Controlled Subjects with Type 2 Diabetes

**DOI:** 10.3390/biom10040520

**Published:** 2020-03-30

**Authors:** Karim G. Kheniser, Abdullah Osme, Chunki Kim, Serguei Ilchenko, Takhar Kasumov, Sangeeta R. Kashyap

**Affiliations:** 1Department of Endocrinology and Metabolism, Cleveland Clinic, 9500 Euclid Avenue, Cleveland, OH 44195, USA; khenisk@ccf.org (K.G.K.); kashyas@ccf.org (S.R.K.); 2Department of Pharmaceutical Sciences, Northeast Ohio Medical University, 4209 St. R., PO Box 95, Rootstown, OH 44272, USA; Abdullah.Osme@case.edu (A.O.); chunki.kim@fiu.edu (C.K.); silchenko@neomed.edu (S.I.)

**Keywords:** HDL dysfunction, heavy water, proteomics, type 2 diabetes, apolipoproteins

## Abstract

We examined the effect of mild hyperglycemia on high-density lipoprotein (HDL) metabolism and kinetics in diet-controlled subjects with type 2 diabetes (T2D). ^2^H_2_O-labeling coupled with mass spectrometry was applied to quantify HDL cholesterol turnover and HDL proteome dynamics in subjects with T2D (n = 9) and age- and BMI-matched healthy controls (n = 8). The activities of lecithin–cholesterol acyltransferase (LCAT), cholesterol ester transfer protein (CETP), and the proinflammatory index of HDL were quantified. Plasma adiponectin levels were reduced in subjects with T2D, which was directly associated with suppressed ABCA1-dependent cholesterol efflux capacity of HDL. The fractional catabolic rates of HDL cholesterol, apolipoprotein A-II (ApoA-II), ApoJ, ApoA-IV, transthyretin, complement C3, and vitamin D-binding protein (all *p* < 0.05) were increased in subjects with T2D. Despite increased HDL flux of acute-phase HDL proteins, there was no change in the proinflammatory index of HDL. Although LCAT and CETP activities were not affected in subjects with T2D, LCAT was inversely associated with blood glucose and CETP was inversely associated with plasma adiponectin. The degradation rates of ApoA-II and ApoA-IV were correlated with hemoglobin A_1c_. In conclusion, there were in vivo impairments in HDL proteome dynamics and HDL metabolism in diet-controlled patients with T2D.

## 1. Background

Cardiovascular disease (CVD) is one of the major complications of type 2 diabetes (T2D) and the most common causes of mortality in patients with T2D [1]. Patients with T2D are characterized by reduced levels of high-density lipoprotein (HDL) cholesterol, a known risk factor for CVD. However, recent pharmacological approaches aiming to improve HDL cholesterol levels had a negligible impact on CVD risk. For instance, clinical trials that increased HDL levels via niacin treatment failed to improve CVD outcomes [2]. Other trials that investigated the effects of cholesterol ester transfer protein (CETP) inhibitors also did not result in reduced CVD risk [3]. Furthermore, the genetic studies echoed this prevailing sentiment by demonstrating that genetic polymorphism associated with the augmented levels of HDL was not associated with reduced CVD risk [4]. Altogether, the aforementioned studies suggest that HDL functionality and HDL quantity are important determinants of CVD risk because there is a clear interplay between the two [5].

HDLs are a mixture of heterogeneous particles with distinct particle sizes, lipids, and protein compositions that collectively influence their numerous functions: prevention of inflammation and oxidation and promotion of endothelial health, which are often impaired in diabetes [6,7,8,9,10,11]. Moreover, HDL-mediated reverse cholesterol transport (RCT) is an important protective function of HDL. Although several in vitro assays measure different aspects of HDL functionality [12,13,14], currently it is difficult to assess the HDL function in humans in vivo.

HDL dysfunction in patients with diabetes has been linked to altered lipid and protein composition [9,10,15,16]. In addition to the major protein of the HDL, apolipoprotein A-I (ApoA-I), multiple less abundant HDL-related proteins with distinct functions have been identified [17]. These HDL proteins are involved in lipid metabolism, acute-phase response, and innate immunity; the aggregate proteins determine both HDL’s anti-inflammatory and anti-atherogenic properties [18]. For instance, ApoA-II, the second most abundant protein of the HDL, is involved in the regulation of cholesterol efflux, HDL remodeling, and cholesteryl ester uptake via its interactions with lipid transfer proteins and lipases [19]. ApoA-IV affects satiation [20], hepatic gluconeogenesis [21], insulin secretion (incretin-like mechanism) [22], and cholesterol efflux [23]. ApoA-IV also has antioxidant, anti-inflammatory, and anti-atherogenic properties [24]. Paraoxonase 1 (PON1), the major antioxidant protein of the HDL, attenuates oxidative modification of lipoprotein particles and contributes to the anti-atherogenic properties of HDL [25]. Although it is known that HDL functionality is altered by inflammation, oxidative stress, and hyperglycemia, the underlying mechanisms that lead to HDL dysfunction in T2D are not fully understood.

While the quantification of HDL cholesterol concentration is an important clinical metric, it does not fully explain the etiology of HDL dysfunction. In contrast, HDL dynamics provides a breadth of information into its function. In particular, HDL flux determines the efficiency of RCT [26]. More so, kinetic measures are more sensitive to the changes in metabolic pathways than static measurements of HDL concentration. Except for ApoA-I [27,28,29,30], little is known about how T2D influences the dynamics of other HDL proteins. It is also unknown whether the HDL cholesterol flux is affected in early stages of diabetes. Recently, we developed a simple ^2^H_2_O-metabolic labeling-based, non-radioactive method to assess integrative cholesterol flux in the reverse cholesterol transport (RCT) pathway [31] and HDL proteome dynamics in vivo [32]. We applied this method to study the kinetics of ApoA-I, ceruloplasmin, and transferrin, HDL proteins involved in cholesterol, copper, and iron transport, respectively, in patients with diet-controlled T2D [28,33]. In this proof-of-concept study, we expanded the utility of ^2^H_2_O-method to investigate the dynamics of HDL cholesterol and multiple other HDL proteins involved in innate immunity, anti-inflammatory, anti-oxidant, and transport functions of HDL in the early stages of T2D.

## 2. Methods

### 2.1. Study Subjects

The Institutional Review Boards of the Cleveland Clinic and NEOMED reviewed and approved the protocol. All volunteers gave their informed written consent prior to partaking in the study. Healthy individuals were recruited by advertisement and/or targeted search within the electronic medical records at the Cleveland Clinic, Ohio. Patients with T2D were recruited at the endocrinology department of the Cleveland Clinic. All patients were newly-diagnosed, insulin-naive individuals, and were not taking any oral hypoglycemic drugs with an average diabetes duration of 1–3 months. Diabetic patients were diagnosed as defined by the American Diabetes Association’s criteria based on an oral glucose tolerance test (≥200 mg/dL after a 2-h 75-g dextrose challenge) and/or a hemoglobin A_1c_ (HbA_1c_) of ≥6.5%. Since diagnosis, all patients with T2D were advised to adhere to lifestyle modification (carbohydrate-controlled diet and moderate physical activity). Follow-up counseling was provided by endocrinologists. Each potential subject underwent medical screening that included a physical examination and blood chemistry profile.

Individuals were excluded if they engaged in self-reported intensive physical activity in the previous six months; furthermore, the prospective subjects were excluded if they had a history of alcohol and/or drug abuse, if they had a history of smoking, or if they had a history of cardiovascular, renal, hepatic, hypothyroid, or hematological diseases. In addition, subjects were excluded if they were taking any lipid-lowering drugs, β-blockers, or agents known to affect lipid metabolism. For the three days immediately prior to and during the one-week HDL turnover study, a registered dietitian counseled all subjects to consume an isocaloric diet with an estimated macronutrient composition of ~45% carbohydrate, ~35% fat, and ~20% protein. All subjects were also advised to avoid strenuous exercise to prevent any exercise-induced changes in HDL metabolism. Each subject underwent a ^2^H_2_O-based kinetic study as described below.

### 2.2. Kinetic Study

An ^2^H_2_O metabolic labeling approach was used to assess HDL cholesterol turnover and HDL proteome dynamics [28]. Briefly, subjects who fasted overnight consumed 70% ^2^H_2_O in their drinking water (4.0 mL ^2^H_2_O/kg body mass) in five doses (given at 0, 1, 2, 3, and 4 h). Additional maintenance doses (10% of the loading dose/day) were taken each day. Blood samples were obtained at 0, 2, 4, 5, 6, 7, 8, and 10 h after the initial dose of ^2^H_2_O. Subjects reported for the scheduled overnight fasted blood draws after 1, 2, 4, and 7 days of initiation of the kinetic study. Serum samples were immediately processed for the kinetics analyses and plasma was stored at −80 °C.

### 2.3. Analytical Procedures

Fasting plasma glucose, triglycerides, total cholesterol, and HDL cholesterol were measured by enzymatic analysis on an automated platform (Roche Modular Diagnostics). Serum levels of high-sensitivity C-reactive protein (hsCRP) were measured in the clinical chemistry laboratory of the Cleveland Clinic. Total protein content was measured by the bicinchoninic acid (BCA) protein assay method. Serum levels of insulin were measured using standard clinical assays. Lecithin–cholesterol acyltransferase (LCAT) activity was measured using a commercially available assay kit, according to the manufacturer’s instructions (Roar Biomedical, New York, NY, USA) [28]. CETP activity was determined by using a fluorometric assay kit II (BioVision Incorporated, Milpitas, CA). In this assay, the donor molecules containing a fluorescent self-quenched phospholipid is transferred by the CETP to an acceptor molecule. CETP activity was monitored for 90 min (excitation/emission = 480/511 nm). The results were quantified in fluorescence arbitrary units, and then the CETP activity was expressed as pmol/mL/min based on the standard curve run with known amounts of donor molecule. The plasma concentration of total adiponectin including full-length and globular form, was performed using the human adiponectin sandwich enzyme-linked immunosorbent assay kit, according to the manufacturer’s instructions (Invitrogen Corporation, Carlsbad, CA, USA). The optical density was measured at a wavelength of 450 nm using the Spectra Max 340PC microplate reader (Molecular Devices, Sunnyvale, CA, USA) and the results were calculated with the optical density from the standard curve. To assess the functional consequence of diabetes-induced changes in the dynamics and remodeling of HDL proteome, we also measured the proinflammatory index of ApoB-depleted plasma [34], the cell-free assay developed by Navab and colleagues [35]. Briefly, oxidation in ApoB-depleted plasma was initiated with Cu^2+^ and rates of total oxidation quantified with 2′,7′-dichlorodihydrofluorescein in a microtiter plate at 37 °C. Fluorescent emission with a 530 nm wavelength was measured after serial excitation at 485 nm. The peroxidase activity of myeloperoxidase (MPO) in plasma was measured by a spectrophotometer with 3,3′5,5′-tetramethylbenzidine (TMB) as the substrate. MPO catalyzed oxidation of TMB was measured at 650 nm. All assays were performed in duplicates.

#### 2.3.1. Body Water ^2^H_2_O Enrichment Measurement

^2^H-enrichment of body water, in parallel with a set of calibration curve samples containing 0–5% ^2^H_2_O was measured using gas chromatography-mass spectrometry (GC-MS) analysis as described [36]. Briefly, 5 μL of plasma was incubated with 5 μL of 10 M potassium hydroxide, and 5 μL of pure acetone in a 2-mL glass screw-cap GC vial at room temperature for 4 h. One μL of acetone vapor from the headspace was directly injected for GC-MS analysis. The ^2^H-enrichment of acetone was determined using electron impact ionization and selected ion monitoring at *m*/*z* 58 (M_0_), 59 (M_1_), and 60 (M_2_). The regression equation of the calibration curve was used for the calculation of ^2^H_2_O enrichment in biological samples.

#### 2.3.2. High-Density Lipoprotein (HDL) Isolation and Proteome Composition

Anti-HDL polyclonal IgY resin column (GenWay BioTech, San Diego, CA, USA) was used to isolate HDL from serum (30 µL), according to the manufacturer’s instruction. The immunocaptured HDL proteins were eluted with three washes of 500 μL stripping buffer (100 mM glycine HCl, pH 2.5) into a tube containing 60 μL of 1.0 M Tris (pH 8.0) to bring pH to 7.0. Eluted fractions were combined, desalted, and concentrated using a 10-kDa centrifugal molecular weight cutoff filter. The protein concentration of immunocaptured HDL fraction was measured by the BCA protein assay method. The purity of the isolated HDL was assessed using 4–20% SDS-PAGE [28]. HDL proteome composition was determined by LC-MS/MS as described below.

#### 2.3.3. HDL Dynamics Analysis

HDL proteome dynamics was assessed in ApoB-depleted serum using the ^2^H_2_O-metabolic labeling approach [31]. Briefly, after precipitation of ApoB-containing particles (IDL and LDL), a magnesium chloride/dextran sulfate reagent (Stanbio Laboratory) [31], the supernatant containing ApoB-depleted serum was used for the analysis of HDL. HDL proteins were precipitated with 1 mL of cold acetone. The supernatant was used for HDL cholesterol analysis by GC-MS [1]. The protein pellet was used for the analysis of HDL proteins as described [37]. Briefly, disulfide bonds of proteins were reduced with dithiothreitol (DTT) and free thiol groups were alkylated with an excess of 2-iodoacetamide. Proteins were digested and tryptic peptides were analyzed by LC-MS/MS. Mass spectrometry analysis was performed on a Q Exactive Plus (Thermo Fisher Scientific, Waltham, MA, USA) instrument using Xcalibur 2.2 software. The details of the mass spectrometric analysis are provided in the Appendix A.

The data were searched with Mascot software (Matrix Science, Version 2.5.1) against the National Center for Biotechnology Information *Human* reference sequence database (ftp://ftp.ncbi.nih.gov/refseq/). The search was performed using cysteine carbamidomethylation as a fixed modification, and methionine oxidation and lysine and arginine glycation as variable modifications with trypsin as the protease. The mass tolerances for the precursor and product ions were 10-ppm and 0.04 Da, respectively. A Mascot score of >35 was considered significant. Unique peptides were identified using BLAST analysis (http://blast.ncbi.nlm.nih.gov/Blast.cgi) as needed. Proteins were characterized based on multiple unique peptides at 99% confidence and a false discovery rate of 1%. HDL proteins (Appendix A) were cross-referenced against the HDL Proteome Watch Initiative database available at http://homepages.uc.edu/~davidswm/HDLproteome.html.

A custom-built software was used for protein, peptide listing, and to calculate isotopic enrichment (*E*) of the ^2^H atoms incorporated into tryptic peptides at each time-point [38]. The proteins identified by the Mascot search engine were filtered based on distinct peptides, peptide and protein score thresholds, and at least four experiments in which a peptide/protein was consistently identified. The peptides were extracted based on the exact mass of the molecular ion with the isolation window of ±10 ppm and MS/MS spectra. ^2^H-isotope content in tryptic peptides was assessed based on mass isotopomer distribution analysis of the high-resolution full-scan spectra as described [39]. Mass isotopomers are molecules that differ by the presence of different heavy isotopes, resulting in a mass spectrum with a baseline monoisotopic (M_0_) peak followed by distinct heavy isotopomer (M_i_, where i is an integer > 0) peaks. Quantification was performed by integrating each isotopomer of a given chromatographic peak within a defined mass range (10 ppm). Only data corresponding to the unique peptides with a Mascot ion score higher than 35 and the intensity of 10^6^–10^7^ were selected for the kinetic analysis. The mass spectrometry proteomics data were deposited with the ProteomeXchange Consortium [40] via the PRIDE partner repository [41] with the dataset identifier PXD005755.

Fractional catabolic rate (FCR, pool/d) HDL proteins were calculated based on ^2^H-incorporation as described [32]. Briefly, the rate constant (*k*) representing the FCR was determined using a one-compartmental model based on the exponential growth curve fitting of net normalized ^2^H- enrichment values of tryptic peptides (*E*(t)) against time (t) after the removal of statistical outliers within the Prism (GraphPad, La Jolla, CA) software (version 5). A time course of the normalized ^2^H net labeling curve was constructed for several (n = 2^−10^) peptides of each protein. Aggregation of a peptide’s relative ^2^H excess curve into protein curves involves the normalization of peptide labeling at each time point to the plateau labeling, averaging, and fitting into the exponential rise curve.
*E*(t) = *E_as_* (1 − e^−*k*t^)(1)

This equation allows determination of asymptotical normalized total labeling (*E_as_*), and the rate constant (k). Protein half-life (t_½_) was calculated based on the rate constant values:(2)$$t_{½}=\frac{\ln2}{k}$$

Due to the slow turnover of PON1, its FCR was calculated based on the precursor and product relationship using Equation (3) [42]:*FCR = Slope of labeling of PON1 peptide/(N × MPE of plasma water)*(3)

The slope of labeling of PON1 peptides was derived from the linear regression analysis of the net labeling of analyzed peptides at different time-points after ^2^H_2_O administration. In this equation, *N* is the asymptotic number of deuterium atoms incorporated into a peptide, which is estimated by integrating the labeling of intracellular free amino acids that make up a peptide, represents the number of exchangeable hydrogen atoms, and is calculated for each peptide based on its amino acid sequence, as previously described [42].

A similar approach was used to calculate the FCR and half-life of HDL cholesterol. We assumed that HDL levels did not change in the adult subject during the seven day ^2^H_2_O-metabolic labeling study period, and that there was a steady-state flux of HDL. Thus, at steady state, the rate constant represents both the FCR and the fractional synthesis rate (FSR).

The production rate (PR) of HDL cholesterol was calculated as the product of their FCR and pool size (mg/kg):*PR (g × kg^−1^ × h^−1^) = pool size × FCR*(4)

The pool size (absolute content) in circulation is the product of HDL cholesterol concentrations and plasma volume, estimated as 4.5% of the body weight.

### 2.4. Data Presentation and Statistical Analysis

The average of the duplicate GC-MS and LC-MS injections, which differed less than 2%, were used for mass-spectrometric analysis. The regression coefficient (R^2^) of the rate constant was used to assess the goodness of the exponential curve fitting. Peptides with a R^2^ value of less than 0.95 were excluded from the data analysis.

The mean and standard deviation were used as descriptive measures for the kinetic measurements; thus, the unpaired two-sided t-test quantified group differences [43,44]. The median was used as a measure of central tendency in Table 1. [43]. When comparing the distribution curves between groups (i.e., control vs. T2D group), Yuen’s method was used along with the percentile bootstrap [45,46]. Finally, for correlation and univariate regression analyses, the percentage bend correlation (*r_pb_*) and the Theil–Sen estimator were used, respectively [47]. For the delineation of confidence intervals (CI), bootstrap methods were employed. The alpha was set at 0.05, and the statistical analysis was conducted in R-studio (version 3.3.1), while the figures were prepared in SPSS (version 23).

## 3. Results

### 3.1. Subject Characteristics

The comprehensive clinical characteristics of the study subjects have been reported in previous publications [28]. Table 1 depicts the basic biochemical and demographic characteristics that are relevant to this report. Subjects with T2D and healthy controls were matched by body weight, BMI, gender distribution, and age without any significant differences in blood pressure and fasting plasma lipids. Patients with T2D were not taking any glucose-lowering agents, instead, the blood glucose levels were controlled by diet. There was a trend toward increased serum levels of hsCRP and MPO activity in patients with T2D, however, these differences were not significant due to high variabilities among subjects. As expected, subjects with diet-controlled T2D had significantly higher levels of HbA1_c_ and blood glucose, and they were insulin resistant as determined based on the homeostatic model assessment (Table 1).

### 3.2. HDL Cholesterol Turnover

We used a single ^2^H_2_O-metabolic labeling approach to study the kinetics of HDL cholesterol and HDL proteins. One hour after the last bolus dose of ^2^H_2_O, ^2^H-labeling of body water was stabilized at 0.80–0.90% in all subjects. ^2^H-labeling of HDL cholesterol increased more rapidly in subjects with T2D than in the controls (Figure 1), suggesting increased turnover. Indeed, subjects with T2D had a significantly higher clearance (FCR) of HDL cholesterol (0.16 ± 0.01 vs. 0.29 ± 0.08 pool/day, *p* = 0.001) compared to the controls (Table 2).

In addition, HDL cholesterol plasma residence times were reduced in subjects with T2D relative to the controls (8.31 ± 3.52 vs. 3.76 ± 1.20 day, *p =* 0.004) (Table 2). Despite increased FCR in subjects with T2D, HDL cholesterol pool sizes were not different. However, the PR of HDL cholesterol was significantly higher in subjects with T2D compared to the healthy controls (Table 2), suggesting that the increase in the PR of HDL compensated for its enhanced clearance, which kept the HDL cholesterol pool size stable.

### 3.3. HDL Proteome Composition and Dynamics

HDL represents a mixture of heterogeneous particles containing exchangeable proteins; the composition of the HDL proteome may be affected by isolation methods [48]. To ensure proteome integrity, we isolated HDL with anti-HDL IgY spin columns and confirmed its purity by using SDS-PAGE. Shotgun proteomics analysis identified 83 overlapping proteins in all subjects (Appendix A). The majority of these proteins have been previously identified as HDL proteins via other isolation methods [49,50]. Although HDL isolation by the immunocapture method preserves the integrity of the HDL proteome, it is costly and cumbersome for routine kinetic applications. To explore the utility of apoB-depletion, a simple method for HDL isolation, we isolated HDL using both dextran sulfate/MgCl_2_ and immunocapture methods. The results demonstrated that the kinetics of the analyzed proteins measured by the two methods were very similar (Appendix A). Since the ApoB-depletion method enabled accurate quantification of the HDL proteins’ kinetics along with HDL turnover, we selected this simple method for further analyses.

Recently, we demonstrated that despite no changes in ApoA-I, ceruloplasmin and transferrin levels, the FCR of these proteins were significantly altered in diet-controlled T2D patients [28,33], indicating that the kinetic measurements were more sensitive than static snapshots of proteins levels. Consistent with these results, we found that the kinetics of several other HDL proteins were also perturbed in the subjects with T2D as compared to the controls (Table 3). In particular, the half-lives of three proteins involved in lipid metabolism were significantly reduced: ApoA-II, ApoJ, and ApoA-IV. Similarly, reductions were observed in the half-lives of transthyretin, vitamin D-binding protein, and complement C3 (C3). In contrast, the half-life of the antioxidant PON1 was significantly increased (Table 3). This result in PON1 kinetics is consistent with our previous report showing reduced PON1 levels and activity in these subjects with T2D [28].

Regression analysis showed that the FCR of a few HDL proteins were directly associated with HbA_1c_ in all studied subjects. In particular, for each one-point increase in HbA1_c_ (x-variable), there was a corresponding 0.005 (CI 0.0020, 0.009) increase in the FCR of ApoA-II (y-variable) (Figure 2A). HbA1_c_ levels also predicted hyperglycemia-associated increased degradation (FCR) of ApoA-IV (Figure 2B).

### 3.4. Adiponectin Levels, LCAT and, CETP Activities, and Pro-inflammatory Index of HDL

Recently, we demonstrated that cholesterol efflux and anti-oxidant properties of HDL were impaired in diet-controlled T2D patients [28]. Adiponectin, LCAT, and CETP may affect HDL functions through altered HDL metabolism, maturation, and remodeling, respectively. To investigate the mechanisms of HDL dysfunction and increased degradation of HDL proteins in these patients, we quantified serum adiponectin levels and LCAT and CETP activities, and their relationship to HDL metabolism and function. Since our HDL proteome dynamics studies revealed that T2D resulted in alterations of the kinetics of acute phase response and antioxidant proteins, we also analyzed the proinflammatory index of HDL in apoB-depleted plasma. This assay determines the capacity of HDL to inhibit or aggravate pro-oxidant -induced oxidation and it is mechanistically linked to oxidative stress. Consistent with a previous report [51], adiponectin levels were lower in the T2D group (Table 1). Although CETP activity was not affected in diabetic subjects (Table 1), it was inversely associated with adiponectin levels in the T2D group (Figure 3).

Similarly, no diabetes-induced changes were observed in the activity of LCAT nor the pro-inflammatory index of HDL (Table 1). A correlative analysis indicated that LCAT activity was inversely associated with fasting blood glucose levels (r_pb_ = −0.84, *p =* 0.036, CI [−1.0, −0.17]) (Table 4). While the pro-inflammatory index of HDL tends to be negatively correlated with HDL cholesterol levels, the association did not reach significance (*p =* 0.07) (Table 4).

Regression analysis determined that T2D-induced alterations in the turnover rates of key HDL proteins were associated with the parameters of HDL maturation and function. Particularly, the pro-inflammatory index of HDL was strongly associated with ApoA-II flux (*p =* 0.005). Additionally, LCAT activity was directly associated with the PON1 turnover rate (*p =* 0.04) (Table 5).

## 4. Discussion

This study used a ^2^H_2_O-labeling approach to measure the turnover rates of HDL cholesterol and HDL proteins in patients with diet-controlled T2DM. The results show that the FCR of HDL cholesterol and several HDL proteins including ApoA-II, ApoJ, and ApoA-IV were increased in patients with diet-controlled T2D. These changes were associated with lower levels of adiponectin, an adipose-tissue-derived peptide that was negatively related to CETP activity. The study extends upon our prior research [28,33] and demonstrates that mild hyperglycemia alters HDL dynamics in the early stages of diabetes.

Patients with diabetes are characterized by reduced HDL cholesterol levels and altered HDL proteome composition [52,53]. Variations in HDL lipid and protein concentrations could be due to either an accelerated synthesis without a measurable change in clearance or a decreased clearance without a change in synthesis. Recently, we developed a ^2^H_2_O-metabolic labeling approach to simultaneously assess the turnover rates of HDL proteins and HDL cholesterol in mice [31,32]. In contrast to other tracers, ^2^H_2_O rapidly equilibrates with the total body water and the intracellular precursors of cholesterol and proteins. Since ^2^H_2_O is administered in drinking water, it is possible to study HDL metabolism in free-living subjects without the need for inpatient intravenous tracer infusion. When we applied this method to diet-controlled subjects with T2D, we found that hyperglycemia resulted in increased ApoA-I degradation, the principal HDL protein largely responsible for reverse cholesterol transport [28]. These changes coincided with a reduced cholesterol efflux capacity of HDL, and a marked shift toward the lipid-poor pre-β HDL particles in patients with T2D. The current study expands on these findings and suggests that the hyperglycemia-induced alterations also affect the stability of other HDL proteins in diet-controlled patients with T2D.

An interesting finding was the 40% increase in vitamin D-binding protein FCR. In addition to being the carrier protein for vitamin D and its metabolites, it is implicated in the pathogenesis of T2D. In particular, vitamin D-binding protein is involved in the inflammatory process because it activates the T-cell response that may lead to the development of type 1 diabetes (T1D) and T2D [54]. Diabetes-induced changes in vitamin D-binding protein turnover could be used to evaluate the effect of vitamin D supplementation on the prevention or treatment diabetes. This is especially important because the metabolite of vitamin D and vitamin D-binding protein do not allow for assessment of the effect of vitamin D on diabetes.

In addition, we found increased turnover rates of C3, the HDL protein involved in complement activation in the early stages of diabetes. Complement activation has been implicated in atherogenesis [55,56,57]. C3 plays an important initiating role in the activation of a complement pathway involved in inflammatory response on the vascular endothelium [58]. It has been shown that increased HDL-bound C3 levels are increased in subjects with T1D and CVD [59]. Enrichment of HDL with C3 has been implicated in suppressed HDL cholesterol efflux capacity [60]. T2D also led to marginal but non-significant alterations in Apo-J (clusterin) kinetics. Low clusterin levels in HDL were associated with insulin resistance and may play a role in the loss of HDL’s protective functions [56]. Although we have not quantified the levels of these HDL proteins, their altered turnover rates may contribute to their altered levels.

Our proteome dynamics study also revealed that the turnover rate of PON1, an antioxidant and associate enzyme of HDLs, was lower in subjects with T2D compared to the healthy controls (Table 3). Interestingly, PON1 expression and activity were diminished in subjects with T2D [28]. Collectively, the data suggested that the PR of PON1 was reduced in subjects with T2D, which contributed to systematic oxidative stress and inflammation in diabetes.

Diabetic-induced alterations in HDL proteome dynamics were reflected by cholesterol cargo turnover in HDLs. Despite no changes in the sex-adjusted HDL cholesterol levels in T2D patients, HDL’s PR and FCR were increased, which contributed to an overall increase in HDL cholesterol flux in diet-controlled subjects with T2D. Thus, in the early stages of diabetes, an increased HDL PR may be compensating for diabetes-induced accelerated clearance of HDL. This finding also underscores the superiority of kinetic analysis, which can detect differences in HDL metabolism compared to common clinical static measurements that dismiss these differences based on HDL levels. Furthermore, the results suggest that in the early stages of diabetes, HDL metabolism is altered but not to an extent where it will impact the clinical characteristics. However, continued disease progression in un-controlled diabetes may result in an impaired PR that will not compensate for the higher catabolism of HDL.

The mechanism by which HDL dynamics is altered in diabetes is not well understood. Non-enzymatic modification (glycation) has been shown to reduce protein stability [53,61] via protein cross-linking and changes in peptide charge, HDL protein conformation, is altered [62]. Recently, we demonstrated that increased degradations of two HDL proteins, ApoA-I and transferrin in normolipidemic diet-controlled subjects with T2D are related to their post-translational glycation [28,33]. Additionally, small lipid-poor HDL particles may be prone to cleavage and degradation as a result of glycation [61,63]. Posttranslational modifications (PTMs) of ApoA-I alter its polarity and conformation that result in perturbed function [62] including impaired cholesterol efflux capacity of HDL in subjects with T2D [28]. Similar to ApoA-I and transferrin, other HDL proteins could also be affected by reactive α-dicarbonyl species and glucose-mediated modifications. Most notably, the FCRs of ApoA-II and ApoA-IV were increased and were positively associated with HbA1_c_, suggesting that hyperglycemia could contribute to HDL dysfunctions due to reduced stability of other HDL proteins. Interestingly, pharmacological agents (e.g., metformin) that inhibit glycation of HDL proteins, also improve HDL functionality [53]. Although the PTM analysis revealed several glycation sites of the analyzed HDL proteins, we failed to quantify the kinetics of glycated proteoforms due to their low abundances; therefore, this area deserves additional investigation.

Consistent with previous studies [64,65,66], we found that plasma adiponectin levels were reduced in diet-controlled patients with T2D. Adiponectin may impact HDL metabolism via changes in HDL composition and particle sizes. HDL’s enrichment with triglycerides and depletion in cholesterol ester correlates with their antioxidative and vasodilatory functions. It has been shown that adiponectin increases cholesterol efflux by increasing the ATP-binding cassette transporter and ApoA-I synthesis [64,65]. Moreover, adiponectin concentration was shown to be negatively correlated with ApoA-I FCR independent of other confounding variables, suggesting that adiponectin is involved in altered ApoA-1 metabolism [51,66]. In the present study, adiponectin was negatively associated with CETP activity. Therefore, we speculate that the adiponectin-mediated changes in HDL lipid composition induced clearance of HDL. While intriguing, additional evidence is needed before a plausible physiological explanation can be determined.

This kinetic human study has several limitations. The study did not elucidate the effect of altered HDL proteome dynamics on concentrations of HDL proteins, and the role of glycation-related PTM on the kinetics of HDL proteins was not discerned. In addition, the sample size was minuscule, and therefore additional studies with a larger number of subjects are warranted. Additionally, studies including lean subjects, as opposed to overweight subjects in this study, may reveal more robust differences between groups and provide additional information on the role of insulin resistance and hyperglycemia during T2D initiation and progression. Nevertheless, this proof-of-concept study demonstrated that the ^2^H_2_O-based metabolic labeling approach enables assessment of both the dynamics of HDL proteins and HDL cholesterol turnover in free-living subjects and detects diabetes-induced changes in HDL metabolism.

## 5. Conclusions

The same factors (e.g., oxidative stress, inflammation, hyperglycemia, etc.) that contribute to T2D progression also affect HDL composition and function. In this study, using a ^2^H_2_O-metabolic labeling approach, we determined that the kinetic profile of multiple HDL proteins and HDL cholesterol were altered in subjects with diet-controlled T2D. This study suggests that altered HDL proteome dynamics may contribute to altered HDL proteome distribution and impaired HDL property. As such, an aberrant kinetic profile may serve as a proxy for impaired HDL function and particle distribution. Thus, ^2^H_2_O-metabolic labeling could be used to understand alterations in both HDL composition and dysfunction in diabetes and other metabolic diseases.

## Figures and Tables

**Figure 1 biomolecules-10-00520-f001:**
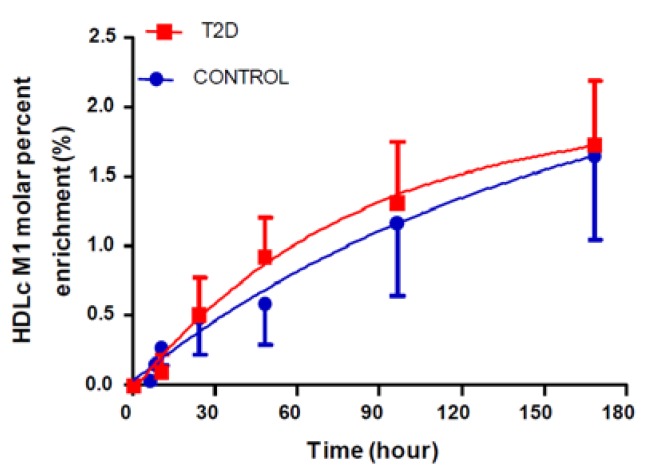
High-density lipoprotein cholesterol turnover in T2D patients (n = 9) and age- and BMI-matched healthy controls (n = 8) determined using a ^2^H_2_O-metabolic labeling approach. A steady-state body water enrichment (~0.8–0.9%) results in gradual labeling of HDL cholesterol, which enables quantification of its turnover rate. Data presented as mean ± SD.

**Figure 2 biomolecules-10-00520-f002:**
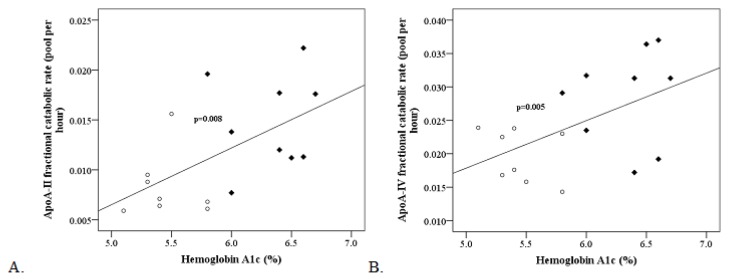
Association of hyperglycemia with the turnover rates of Apo-II and ApoA-IV. As hemoglobin A1c levels are increased, ApoA-II and ApoA-IV half-lives are decreased. Panel (**A**): Hemoglobin A1c levels and ApoA-II fractional catabolic rates are strongly correlated (slope = 0.005 [CI 0.0020, 0.009]). Panel (**B**): Hemoglobin A1c levels predict ApoA-IV kinetics (slope = 0.008 [CI -8.7e^−05^, 0.013]). Open circles and close rhombus signs represent healthy controls and individuals with type II diabetes, respectively.

**Figure 3 biomolecules-10-00520-f003:**
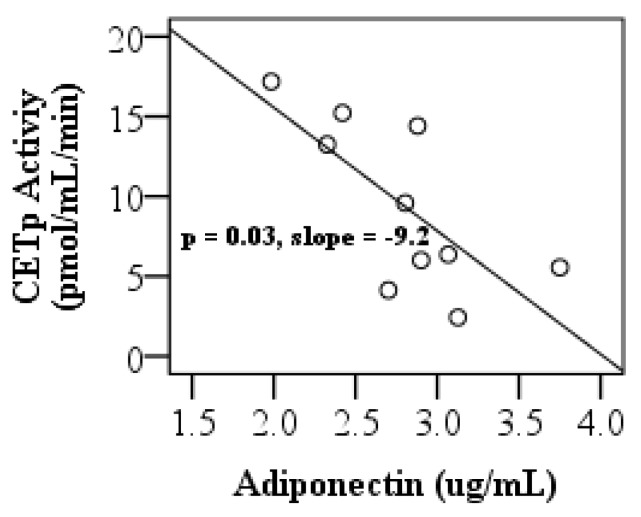
The negative relationship between adiponectin and cholesteryl ester transfer protein activity. Only in the type II diabetes group, adiponectin predicted cholesteryl ester transfer protein activity (slope CI [−15.9, −1.8]).

**Table 1 biomolecules-10-00520-t001:** Demographic and biochemical characteristics of study subjects.

Variables	T2D Median	(min, max)	Control Median	(min, max)	*p*-Value
Number (M:F)	8(4:4)		9(4:5)		
Age (years)	62	(42, 71)	55	(35, 66)	0.072
Weight (kg)	91.8	(76, 121)	82	(57.1, 102.4)	0.175
Body mass index (kg/m^2^)	32.2	(26, 36)	30	(24, 32)	0.175
Systolic blood pressure (mmHg)	131	(101, 158)	123	(99, 143)	0.16
Diastolic blood pressure (mmHg)	72	(55, 97)	77	(60, 90)	0.69
Glucose (mg/dL)	110	(91, 143)	96.5	(78, 108)	0.001
Fasting insulin (mIU/dL)	16.21	(12, 39)	7.0	(4.1, 16.5)	<0.001
HOMA IR	5.2	(3, 10)	1.6	(1, 4)	<0.001
HbA1_c_ (%)	6.4	(5.8, 6.7)	5.4	(5.1, 5.8)	<0.001
Triglycerides (mg/dL)	94	(28, 149)	85	(50, 173)	0.64
High-sensitivity C-reactive protein (mg/L)	2.5	(0.5, 15)	1.1	(0.2, 3.6)	0.253
Myeloperoxidase activity (mOD/µL/min)	7.8	(6.2, 24.1)	6.0	(3.3, 17.8)	0.720
Total cholesterol (mg/dL)	195	(122, 237)	180.5	(110, 254)	0.44
HDL-cholesterol (mg/dL)	48	(33, 67)	52	(34, 79)	0.75
Lecithin-cholesterol acyltransferase activity	1.14	(1, 1.2)	1.1	(1, 1.2)	0.62
Cholesteryl ester transfer protein activity (pmol/mL/min)	11.4	(5.6, 17.2)	9.7	(5.6, 13.6)	0.82
Proinflammatory index of HDL (RFU mg HDLc/min)	19.6	(13.4, 31)	19.9	(12.9, 25.4)	0.87
Adiponectin (µg/mL)	2.9	(1.9, 3.8)	9.5	(2.7, 13)	<0.001

Yuen’s method, T2D: type 2 diabetes, HDL: high-density lipoproteins, HOMA: homeostatic model assessment.

**Table 2 biomolecules-10-00520-t002:** High-density lipoprotein (HDL) cholesterol kinetic parameters in subjects with T2D (n = 9) and age- and body mass index (BMI)-matched healthy controls (n = 8).

Variables	Control	T2D	*p*-Value
HDL cholesterol pool (mg/kg)	23.5 ± 6.1	23.9 ± 7.4	NS
HDL cholesterol FCR (day^−1^)	0.16 ± 0.01	0.29 ± 0.08	0.008
HDL cholesterol residence time (day)	8.31 ± 3.52	3.76 ± 1.20	0.004
HDL cholesterol PR (mg/kg/d)	3.5 ± 1.8	7.2 ± 3.2	0.015

Unpaired two-sided t-test, HDL: high-density lipoproteins, FCR: fractional catabolic rate, PR: production rate.

**Table 3 biomolecules-10-00520-t003:** Half-lives of HDL proteins in healthy controls (n = 8) and subjects with T2D (n = 9).

Accession Number	Protein	GO-Molecular Function	Control t ½ (hour ± SD)	T2D t ½ (hour ± SD)	*p*-Value (95% CI)
**Immune Response**					
P01024	Complement C3	Endopeptidase inhibitor	54.5 ± 17.8	33.3 ± 13	0.002 (−49, −7)
P10909	Apolipoprotein J (Clusterin)	Misfolded protein binding	26.5 ± 10.6	19.8 ± 5.4	0.007 (−19, −2)
**Antioxidant**					
P27169	Paraoxonase 1	Antioxidant	261.6 ± 66.0	474.8 ± 1001	0.006 (64, 301)
**Lipid Metabolism**					
P02652	Apolipoprotein A-II	Cholesterol transporter	91.9 ± 23.1	51.9 ± 17.3	0.003 (−59, 14)
P02656	Apolipoprotein C-III	Lipid-binding	24 ± 5.8	21.9 ± 6.3	0.6 (−9,4)
P06727	Apolipoprotein A-IV	Antioxidant, cholesterol transporter	36.5 ± 7.2	26 ± 7.2	0.004 (−19, −4)
P02766	Transthyretin	Thyroid hormone-binding protein	37.1 ± 12.9	28.3 ± 8.3	0.03 (−27, −1)
**Vitamin Transport**					
P02774	Vitamin D-binding protein	Actin-binding, vitamin transporter activity	50.8 ± 14.2	30.8 ± 8.3	0.02 (−25, −1)

Yuen’s method, CI: Confidence interval. The kinetics of HDL proteins were quantified based on unique peptides. Unique peptides were confirmed using BLAST analysis (http://blast.ncbi.nlm.nih.gov/Blast.cgi).

**Table 4 biomolecules-10-00520-t004:** Relationships between HDL parameters and T2D variables.

Parameters	Lower CI	Upper CI	Correlation (r_pb_)	*p*-Value
LCAT activity Blood glucose concentration	−1.0	−0.17	−0.84	**0.036**
Proinflammatory HDL	−0.98	0.08	−0.75	0.07
HDL concentration				

Percentage bend correlation. CI: confidence interval, LCAT: lecithin-cholesterol acyltransferase, HDL: high-density lipoprotein.

**Table 5 biomolecules-10-00520-t005:** Regression analysis (T2D group).

Parameters	Slope	Slope CI	Intercept	*p*-Value
Proinflammatory HDL (x-variable)	−0.0008	(−0.0013, −0.0003)	0.03	**0.005**
ApoA-II kinetics (y-variable)				
LCAT activity (x-variable)	−0.005	(−0.012, −0.0018)	0.007	**0.04**
Paraoxonase 1 kinetics (y-variable)				

Theil–Sen estimator. ApoA-II: apolipoprotein A-II, CI: confidence interval, LCAT: lecithin-cholesterol acyltransferase, HDL: high-density lipoprotein.

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
