# Peer review of "Temporal Dynamics of High-Density Lipoprotein Proteome in Diet-Controlled Subjects with Type 2 Diabetes"

_biomolecules, 2020, doi:10.3390/biom10040520_

Round 1

Reviewer 1 Report

The paper is potentially interesting since it focus on the use of 2H2O-labeling approach to measure the turnover rates of HDL cholesterol and HDL proteins in patients with diet-controlled T2DM. The topic is interesting and scientifically important, but there are several points need to be explained and revised before it is acceptable.

Some concerns are about the statistics. Please, show results in table 1 as media ± SD, rather than SE to better evaluate the variability of samples in the group. However, the use of median (min; max) should be more robust with respect to outliers.

Although the authors reports their previous references on clinical characteristics of the study population, it should be better include other features, like sex, CRP levels or other inflammatory markers.

It is not clear what variables constitute potential confounders. How about therapy and diet?

In diabetic group, glucose levels are 112mg/dl. Although the authors refer that diabetic patients were diagnosed as defined by the American Diabetes Association’s criteria based on HbA1c and glucose levels after oral glucose tolerance test, it could be useful to explain if glucose levels (112mg/dl) depend on therapy.

Author Response

Please see the word document.

Reviewer 2 Report

The authors present a novel study utilizing an H2O metabolic labeling approach to assess variations in HDL metabolism and kinetics in healthy versus T2DM participants. 

  • there is a very small n, authors address in this in the discussion, but it is concerning. Although significant findings were identified in the current study further investigations with more participants is warranted. 
  • the "healthy" controls had slightly elevated blood glucose values and borderline A1C, most likely due to elevated BMI, was BMI-matching between groups intended? Perhaps investigating variations between T2DM and lean individuals would provide more pronounced results. This should be added to the discussion. 

Author Response

Please see the word document.

Round 2

Reviewer 1 Report

Thank you for your revision of this manuscript. I appreciate the time that you have put into revising your manuscript based upon my comments.

Reviewer 2 Report

Thank you for addressing the concerns raised in the previous review.